# SPINT: Spatial Permutation-Invariant Neural Transformer for Consistent Intracortical Motor Decoding

**Trung Le**[1,4]**, Hao Fang**[1]**, Jingyuan Li**[1]**, Tung Nguyen**[2]**,**
**Lu Mi**[3]**, Amy Orsborn**[1]**, Uygar Sümbül**[4]**, Eli Shlizerman**[1,*]
[1] University of Washington
[2] University of California, Los Angeles
[3] Tsinghua University
[4] Allen Institute

## Abstract

Intracortical Brain-Computer Interfaces (iBCI) decode behavior from neural population activity to restore motor functions and communication abilities in individuals with motor impairments. A central challenge for long-term iBCI deployment is the nonstationarity of neural recordings, where the composition and tuning profiles of the recorded populations are unstable across recording sessions. Existing approaches attempt to address this issue by explicit alignment techniques; however, they rely on fixed neural identities and require test-time labels or parameter updates, limiting their generalization across sessions and imposing additional computational burden during deployment. In this work, we address the problem of cross-session nonstationarity in long-term iBCI systems and introduce SPINT - a **S**patial **P**ermutation-**I**nvariant **N**eural **T**ransformer framework for behavioral decoding that operates directly on unordered sets of neural units. Central to our approach is a novel *context-dependent positional embedding* scheme that dynamically infers unit-specific identities, enabling flexible generalization across recording sessions. SPINT supports inference on variable-size populations and allows few-shot, gradient-free adaptation using a small amount of unlabeled data from the test session. We evaluate SPINT on three multi-session datasets from the FALCON Benchmark, covering continuous motor decoding tasks in human and non-human primates. SPINT demonstrates robust cross-session generalization, outperforming existing zero-shot and few-shot unsupervised baselines while eliminating the need for test-time alignment and fine-tuning. Our work contributes an initial step toward a robust and scalable neural decoding framework for long-term iBCI applications.

## 1 Introduction

Motor behavior arises from the complex interplay between interconnected neurons, each possessing distinct functional properties [1]. Deciphering the highly nonlinear mapping from the activity of these neural populations to behavior has been a major focus of intracortical Brain-Computer Interfaces (iBCI), whose applications have enabled individuals with motor impairments to control external devices [2, 3], restore communication abilities through typing [4, 5], handwriting [6, 7], and speech [8, 9, 10, 11, 12].

Despite the remarkable capabilities, iBCI systems suffer from performance degradation over extended periods of time, largely attributed to the nonstationarities of the recorded populations [13]. Sources of

---

[*]Corresponding author: shlizee@uw.edu. Code is available at `https://github.com/shlizee/SPINT`

nonstationarities include shifts in electrode position, tissue impedance changes, and neural plasticity [14, 15, 16]. These nonstationarities lead to changes in the number and identity of neural units picked up by recording electrodes over time. Such changes in population composition alter the learned mapping from neural activity to behavior, preventing decoders trained on previous sessions from maintaining robust performance in new sessions. To ensure robust behavior decoding over future recording sessions, one approach has focused on training deep networks using many sessions, attempting to achieve decoders that are robust to cross-session variability [17, 18]. This zero-shot approach requires months of labeled training data, necessitating extensive data collection from the user. While recent methods targeting cross-subject generalization may alleviate some of this burden [19, 20, 21, 22], degradation over long-term use still remains, advocating for the adoption of adaptive methods [14, 23, 24, 25, 26]. These adaptive methods leverage the low-dimensional manifold underlying population activity, which has been shown to preserve a consistent relationship with behavior over long periods of time [14, 27]. Depending on the use of labels at test time, they can be categorized into supervised [21, 28, 22], semi-supervised [7], or unsupervised [24, 25, 29], with varying levels of success and practical utility in real-world iBCI [26].

Despite the variety of technical approaches, these works share a common design philosophy: they adopt a fixed view of the neural population, assigning fixed identities and order for neural units during training. While this treatment achieves high decoding performance on held-in sessions (within-session generalization), the decoders suffer from out-of-distribution performance degradation when evaluated on held-out sessions with different sizes and unit membership (cross-session generalization). To enable transfer of the pretrained model to novel sessions, explicit alignment procedures with gradient updates to adapt model parameters are necessary, imposing disruptive and costly computation overhead for iBCI users. With these limitations of existing approaches, we advocate for the view that an ideal, universal iBCI decoder should be invariant to the permutation of the neural population by design, and should be able to seamlessly handle inference of a variable-sized, unordered set of neural units with minimal data collected from the new setting.

In this work, we address the problem of cross-session nonstationarity in long-term iBCI systems and introduce SPINT - a permutation-invariant framework that can decode motor behavior from the activity of unordered sets of neural units. We contribute toward an iBCI design that can predict behavior covariates from continuous streams of neural observations and adapt gradient-free to novel sessions with few-shot unlabeled calibration data. At the core of our methods is a permutation-invariant transformer with a novel context-dependent positional embedding that allows flexible identification of neural unit identities on-the-fly. We evaluate our approach on three movement decoding datasets from the FALCON Benchmark [26], demonstrating robust cross-session generalization on motor tasks in human and non-human primates. Our model outperforms zero-shot and few-shot unsupervised baselines, while not requiring any retraining or fine-tuning overhead.

In summary, the contributions of this work include:

- We present a *transformer-based permutation-invariant framework* with a novel *context-dependent positional embedding* for few-shot unsupervised behavioral decoding. Our flexible, lightweight model enables ingestion of unordered sets of neural units during training and facilitates out-of-the-box inference on unseen neural populations.

- We evaluate our model on three motor behavioral decoding datasets in the FALCON Benchmark, showing robust gradient-free generalization to unseen sessions in the presence of cross-session nonstationarities.

## 2   Related Work

**Decoding motor behavior from intracortical population activity**: A variety of computational models have been proposed to model population spiking dynamics and decode motor behavior, starting with linear models such as population vectors [1], linear regression [30], Kalman filter [31, 32, 33]. Nonlinear methods have also been developed; a non-exhaustive list includes generalized linear models [34, 35, 36, 37], latent variable models [38, 39, 40, 41], leveraging deep neural networks including recurrent neural networks (RNN) [27, 42, 43] and transformers [44, 21, 28, 45, 46, 22, 47, 48, 49]. Most of these works assume a fixed view of the population across time and have limited direct transferability to unseen populations. We adopt the view of neural population as an unordered set of neural units, offering more flexibility in transferring across population compositions.

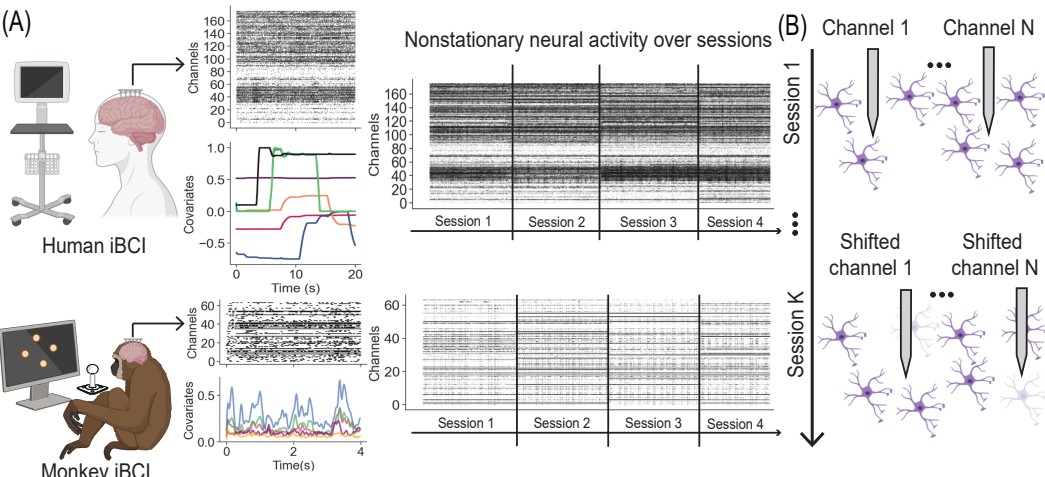

Figure 1: **Nonstationarities in long-term iBCI.** (A) Examples of iBCI systems in human and non-human primates. Spiking activity is recorded from multichannel electrode arrays together with behavior covariates, e.g., 7 degree-of-freedom robotic arm control or electromyography from the upper limb. Neural activity exhibits nonstationarities over recording sessions. (B) Systematic changes in neuron positions, including the introduction or loss of neurons in the vicinity of electrodes and the shifts of the entire electrode array can contribute to instability of neural recordings over time. Illustrations were created with BioRender.com.

**Cross-session neural alignment methods**: Neural nonstationarities pose challenges for maintaining decoder performance over long periods of time. To tackle this issue, alignment techniques were proposed to align the testing sessions to match the distribution of a training session, usually performed in the latent space. These techniques vary from linear methods using Canonical Correlation Analysis (CCA) [14, 20], linear stabilizer [23], linear distribution alignment [24], to nonlinear using generative adversarial networks (GAN) [50, 29], RNN [25, 7, 51], and diffusion models [52]. All these approaches, however, still require explicit alignment procedures with model parameter updates and test-time labels in some cases to adapt the pretrained decoder to unseen populations. On the other hand, our proposed context-dependent positional embedding scheme allows flexible, gradient-free adaptation in new sessions without any labels.

**Foundation models for neuroscience**: Recent advances in foundation models for neural decoding demonstrate promising transferability across sessions, subjects and tasks through pretraining on large and diverse corpora of neural recordings [21, 28, 22, 47, 48, 53, 49]. While these approaches aim to build generic pretrained models that are then adapted to diverse downstream settings via parameter finetuning, SPINT instead focuses on reducing calibration burden in real-world iBCI systems by achieving gradient-free cross-session generalization under the limited data regime. SPINT, however, also ingests data from multi-session recordings during training to achieve this goal. Architecturally, SPINT employs the cross-attention mechanism to integrate information across a variable number of tokens similar to POYO-style models [22, 47, 48], while also operates on binned spike counts similar to NDT-style frameworks [21, 28, 53, 49] and POYO+ in spirit [47]. Unlike these models, SPINT adopts a much shallower design—using only a single cross-attention layer—and tokenizes data by treating each neural unit's temporal context window as a token, analogous to the spatial tokenization in [46, 54], and distinct from the temporal tokenization in [44, 55], patch-based tokenization in [21], or spike-based tokenization in [22, 48].

**Permutation-invariant neural networks for set-structured inputs**: While conventional neural networks are designed for fixed dimensional data instances, in many set-structured applications such as point cloud object recognition or image tagging, the inputs have no intrinsic ordering, advocating for a class of models that are permutation-invariant by design [56, 57]. One such work, DeepSets, introduced a set average pooling approach serving as a universal approximator for any set function [56]. Follow-up works [57, 58] extended this pooling method to include max-pooling and attention mechanisms [59]. We take inspiration from these works to design our universal neural unit identifier and permutation-invariant behavior decoding framework, leveraging the permutation-invariant property of the cross-attention mechanism [60].

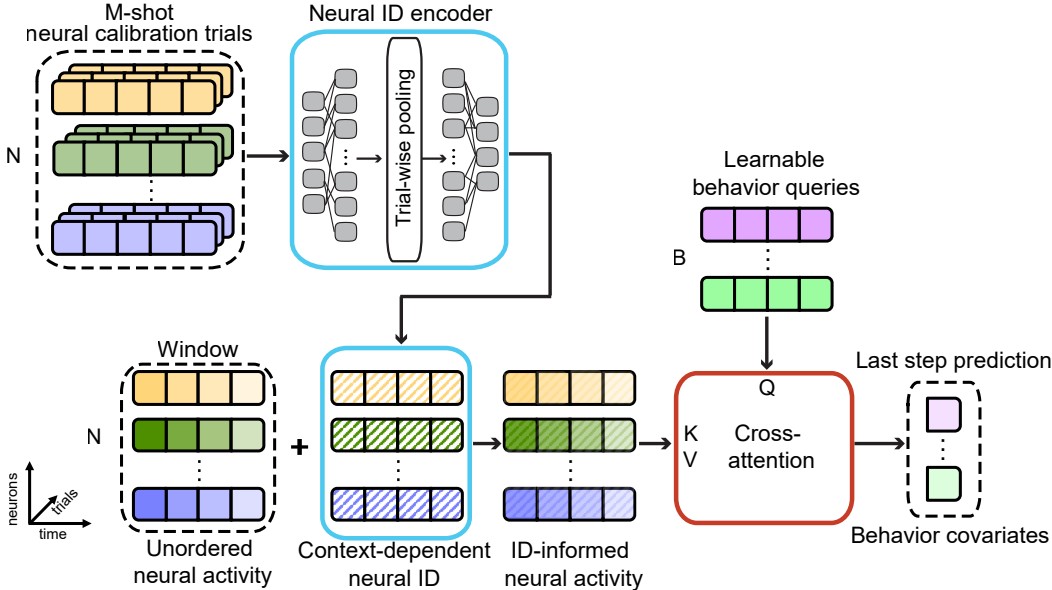

Figure 2: **SPINT architecture.** The model performs continuous behavioral decoding by predicting behavior covariates at the last timestep given a past window of activity from an unordered set of neural units. The universal Neural ID Encoder infers identities of the units using few-shot unlabeled calibration trials, while the cross-attention mechanism selectively aggregates information from the units to decode behavior.

## 3 Approach

### 3.1 A permutation-invariant framework for few-shot continuous behavioral decoding

We study the problem of real-time, cross-session iBCI decoding, where behavior needs to be decoded in a causal manner from a continuous stream of neural observations. Concretely, within a single session $s$, let $X_{i,t}$ denote the binned spiking activity of neural unit $i$ at time $t$, $X_{i,:}$ denote all the activity of unit $i$, and $X_{:,t}$ denote the activities of all units at time $t$. Given a past observation window of population activity $X_{:,t-W+1:t} \in \mathbb{R}^{N_s \times W}$, where $N_s$ is the number of recorded neural units in session $s$ and $W$ is the length of the observation window, we aim to estimate the corresponding last time step of behavior output $Y_t \in \mathbb{R}^B$, where $B$ is the dimensionality of behavior covariates. Model parameters are fitted with gradient descent using labeled data from $k$ training (held-in) sessions and evaluated on $k'$ testing (held-out) sessions without gradient updates or labels. At our disposal on each held-out session is a short calibration period $X_{i,[C]} \in \mathbb{R}^{T'}$ consisting of $M$-shot variable-length trials lasting for $T'$ timesteps, to be used for cross-session adaptation.

Traditional approaches consider the population activity at each timestep as a "token" and decode behavior by modeling temporal dynamics of population activity [39, 27, 44]. By treating temporal snapshots of population activity $X_{:,t} \in \mathbb{R}^{N_s}$ as input vectors, these approaches assume a fixed number and order of neural units, requiring explicit spatial re-alignment when applied to another session with a different size and order [27, 25, 29, 20, 24]. Recent methods incorporating factorized spatial-temporal modeling [46, 61, 45] face similar challenges, while approaches with explicit spatiotemporal tokens [21, 28, 22, 47, 48] still require fine-tuning unit identity in novel sessions. These design choices hinder out-of-the-box generalizability of neural decoders across sessions, as a universal decoder should ideally be invariant to the permutation and size of the input population.

To realize this goal, we treat windows of individual neural units $X_{i,t-W+1:t} \in \mathbb{R}^W$ as an *unordered set* of tokens and aggregate information from these units to decode behavior using the cross-attention mechanism [59]. To compensate for the loss of consistent order that the decoder can leverage for behavior decoding, we embed a notion of neural identity to each unit based on its spiking signature during a few-shot, *unlabeled* calibration period $X_{:,[C]}$ in the same session. $X_{:,[C]}$ is either provided in limited amount in held-out sessions at test time, or is artificially sampled from held-in sessions

during training. This context-dependent neural identity is inferred by a universal neural identity encoder that is shared across units and sessions, enabling *gradient-free* adaptation to novel population compositions at test time.

## 3.2 Encoding identity of neural units

Let $X_i^C \in \mathbb{R}^{M \times T}$ be the trialized version of $X_{i,[C]} \in \mathbb{R}^{T'}$. $X_i^C$ is the collection of $M$ calibration trials of neural unit $i$ interpolated to a fixed length $T$. We infer neural identity $E_i \in \mathbb{R}^W$ of unit $i$ by a neural network IDEncoder:

$$E_i = \text{IDEncoder}(X_i^C) = \psi(pool(\phi(X_i^C))) \tag{1}$$

where $\psi$ and $\phi$ are multi-layer fully connected networks and *pool* is the mean pooling operation across $M$ trials. Due to the permutation invariant nature of the mean operation and the fact that $\psi$ and $\phi$ are applied trial-wise, IDEncoder is invariant to the order of $M$ calibration trials by design [56].

## 3.3 Decoding behavior via selective aggregation of information from neural population units

After inferring the identity for each unit from its calibration period, we add $E_i$ to all $X_i$ windows to form identity-aware representations $Z_i$. $Z_i$ contains the time-varying activity of each unit while also being informed of the unit's stable identity within one session. In matrix form:

$$Z = X + E \tag{2}$$

where $Z_i$'s, $X_i$'s, $E_i$'s constitute rows of the $Z, X, E$ matrices.

We leverage the cross-attention mechanism to selectively aggregate information from population units and decode behavior outputs:

$$Y = \text{CrossAttn}(Q, Z, Z) = \text{softmax}\left(\frac{QK^\top}{\sqrt{d_k}}\right)V \tag{3}$$

where $K = ZW_K, V = ZW_V \in \mathbb{R}^{N_s \times W}$ are projections of the identity-informed neural activity $Z$, and $Q \in \mathbb{R}^{B \times W}$ is a learnable matrix to query the behavior from Z. Q is initialized at random and learned during training. In future work, SPINT can be extended to accommodate multi-task setup by having multiple Q matrices, one for each behavior task, as in [47]. We use the standard cross-attention module with pre-normalization and feedforward layers (see details in Appendix).

**Proposition 1.** *Cross-attention with identity-informed neural activity (Equation 3) is invariant to the permutation of neural units, i.e.,*

$$\text{CrossAttn}(Q, Z, Z) = \text{CrossAttn}(Q, P_R Z, P_R Z), \tag{4}$$

*where $P_R$ is the row permutation matrix. (See proof in Appendix).*

$E$ in Equations 2 and 3 can be understood as a special kind of positional embedding for attention mechanism, where $E$ is equivariant to the order of tokens (neural units), i.e., permuting the rows in $X$ also permutes the rows in $E$ accordingly. Hence, unlike traditional positional embeddings in the transformer literature where positional embeddings are fixed entities, our proposed $E$ is *context dependent*. This context-dependent positional embedding enables cross-session generalization by design, as $E$ is stable for all samples within the same context (session), and can readily adapt in a gradient-free manner to new populations with arbitrary size and order.

After cross-attention, we project down $Y$ by a fully connected layer to a one-dimensional vector representing the predicted behavior covariates at the last timestep, based on which we compute the mean squared error (MSE) between the predicted and the ground truth behavior covariates. The IDEncoder and cross-attention module are trained in an end-to-end manner using this MSE objective.

### 3.4 Encouraging model robustness to inconsistent population composition

Neural population distributes its computation among many neural units, allowing us to effectively decode behavior even though we can only record neural activity with a limited number of electrodes. Leveraging this insight and in order to encourage model robustness to different compositions of neural membership across recording sessions, we employ *dynamic channel dropout* to avoid overfitting to the population composition seen during training. Unlike classical population dropout methods where only a fixed fraction of neurons/timesteps [62, 63, 44] or a variable fraction of neurons/timesteps sampled from a conservative range [64, 22] is removed during training, we uniformly sample a dropout rate between 0 and 1 each training iteration and remove population units with the sampled dropout rate. With dynamic channel dropout, we not only encourage the model to be robust to the unit membership but also encourage it to be robust to the size of the population, leading to improved cross-session generalization without additional tuning of dropout rates (see Section 4.8 and Figure 4).

### 3.5 Gradient-free, few-shot cross-session adaptation in unseen neural populations with variable size and order

The overall framework is depicted in Figure 2. We use labeled data from all training sessions to train all model parameters following the above pipeline. The model naturally digests populations of arbitrary size and order in all sessions without any need of session-specific alignment layer or fixed positional embeddings for neural units, hence having the potential to scale up to a large amount of data. When testing on a held-out session, we reuse the trained IDEncoder and only need a few *unlabeled* calibration trials to infer identities of neural units in the test session, without the need of gradient descent updates to fine-tune session-specific alignment layers or unit/session embeddings. With these benefits, our proposed model removes the time and computation overhead usually required for re-calibrating neural decoders before each session, and facilitates its applicability in real-world iBCI settings where test-time labels are inherently unavailable.

## 4 Experiments

### 4.1 Datasets and evaluation metrics

We evaluate our approach on three continuous motor decoding tasks from the Few-shot Algorithms for Consistent Neural Decoding (FALCON) Benchmark [26]. Specifically, we evaluate SPINT on the M1, M2, and H1 datasets. In M1, a monkey reached to, grasped, and manipulated an object in a variety of locations (4 possible objects, 8 locations), while neural activity was recorded from precentral gyrus and intramuscular electromyography (EMG) was recorded from 16 muscles [65, 66, 67, 68]. In M2, a monkey made finger movements to control a virtual hand and acquired cued target positions while neural activity from the precentral gyrus and 2-D actuator velocities were captured [69]. In H1, a human subject attempted to reach and grasp with their right hand according to a cued motion for a 7-degree-of-freedom robotic arm control [3, 70, 71]. Each dataset comprises multiple labeled held-in sessions used to train the decoder (spanning 4, 4, and 6 days for M1, M2, and H1, respectively), and multiple held-out sessions for model evaluation (spanning 3, 4, and 7 days for M1, M2, and H1, respectively). Each held-out session provides a few public calibration trials (with optional labels) used for decoder calibration, after which the decoder is evaluated on a private test split. Cross-session performance is quantified by the mean and standard deviation of $R^2$ between the predicted and ground truth behavior covariates across all held-out sessions. All evaluation results were obtained on the held-out private split by submitting models to the EvalAI platform [72].

### 4.2 Baselines

We compare SPINT with zero-shot (ZS) and few-shot unsupervised (FSU) baselines, since SPINT is the intersection of these two approaches. Similar to FSU approaches, SPINT makes use of a few unlabeled calibration samples in the held-out sessions; however, unlike conventional FSU approaches, SPINT does not require gradient updates for model parameters at test time, therefore bearing resemblance to ZS methods in terms of practical utility. We call this new class of model *gradient-free few-shot unsupervised* (GF-FSU).

**ZS Wiener Filter and ZS RNN**: Wiener Filter is a linear model that predicts the current behavior as a weighted sum of previous timesteps [73]. In addition to the Wiener Filter, we also compare with a

| | Class | M1 | M2 | H1 |
|---|---|---|---|---|
| Wiener Filter (WF) | OR | $0.53 \pm 0.04$ | $0.26 \pm 0.03$ | $0.21 \pm 0.04$ |
| RNN | OR | $0.75 \pm 0.05$ | $0.56 \pm 0.04$ | $0.44 \pm 0.13$ |
| NDT2 Multi [21] | OR | $0.78 \pm 0.04$ | $0.58 \pm 0.04$ | $0.63 \pm 0.08$ |
| NDT2 Multi [21] | FSS | $0.59 \pm 0.07$ | $0.43 \pm 0.08$ | $0.52 \pm 0.04$ |
| WF | ZS | $0.34 \pm 0.06$ | $0.06 \pm 0.04$ | $0.16 \pm 0.03$ |
| RNN | ZS | $-0.60 \pm 0.45$ | $-0.07 \pm 0.23$ | $0.09 \pm 0.18$ |
| CycleGAN + WF [29] | FSU | $0.43 \pm 0.04$ | $0.22 \pm 0.06$ | $0.12 \pm 0.06$ |
| NoMAD + WF [25] | FSU | $0.49 \pm 0.03$ | $0.20 \pm 0.10$ | $0.13 \pm 0.10$ |
| **SPINT (Ours)** | GF-FSU | $\mathbf{0.66} \pm 0.07$ | $\mathbf{0.26} \pm 0.13$ | $\mathbf{0.29} \pm 0.15$ |

Table 1: Performance comparison against oracles (OR), few-shot supervised (FSS), few-shot unsupervised (FSU), and zero-shot (ZS) methods. Our SPINT approach belongs to a special class which we termed gradient-free few-shot unsupervised (GF-FSU), where models perform adaptation based on few-shot unlabeled data but without any parameter updates at test time. Results are reported as mean $\pm$ standard deviation $R^2$ across held-out sessions.

simple RNN baseline (implemented as an LSTM [74]). The WF and RNN models were fitted using a single held-in session and evaluated zero-shot on the held-out sessions.

**CycleGAN [29]**: An FSU method where a Generative Adversarial Network (GAN) is trained using calibration data from a held-out session (day K) to transform day K's population activity to a form resembling activity from a held-in session (day 0), allowing decoders pretrained on day 0 to be reused on day K.

**NoMAD [25]**: Another FSU method where a dynamical model and a decoder are trained on day 0 to predict behavior from the inferred dynamics. Then on day K, an alignment network is trained to match the distribution of neural latent states to that of day 0, allowing the fixed model and decoder to transfer to day K.

**Wiener Filter, RNN and Transformer Oracles (OR)**: We include the Wiener Filter, RNN, and NDT2 - a transformer for neural data [21], trained on private held-out labeled data to serve as upper bounds for model performance.

**NDT2 Multi (FSS) [21]**: Similar to NDT2 Multi OR, but only trained on held-in and held-out few-shot calibration data with supervision.

### 4.3 SPINT outperforms zero-shot and few-shot unsupervised baselines on continuous motor decoding tasks

We show in Table 1 the performance of SPINT in comparison with ZS and FSU approaches. SPINT outperforms all ZS and FSU baselines across all three datasets, while requiring no retraining or fine-tuning of model parameters. Improvement is most prominent in M1, where the amount of training data is the largest ($\sim 5\times$ more data than H1 and $\sim 6\times$ more data than M2 in terms of recording time). Notably, SPINT surpasses Wiener Filter oracles in all datasets, which were trained with access to the private labeled data. SPINT even outperforms the FSS method NDT2 Multi on M1 dataset while unlike NDT2, it does not require access to test-time labels or model parameter updates. As we focus on cross-session transferability, all our experimental results show the cross-session performance. We include comparison on within-session performance in the Appendix.

### 4.4 SPINT requires only a minimal amount of unlabeled data for adaptation

To gauge the data efficiency of our model at test time, we trained and tested the model with varying number of calibration trials used to infer the neural unit IDs. We show in Figure 3(A) that SPINT could achieve reasonable cross-session generalization with a small number of few-shot trials. In M1 dataset, the model could even achieve similar performance as the best model (which uses all available calibration trials) with only one single trial. This study demonstrates the practical utility of SPINT in online iBCI, relieving the burden of data collection and label collection on users at test time.

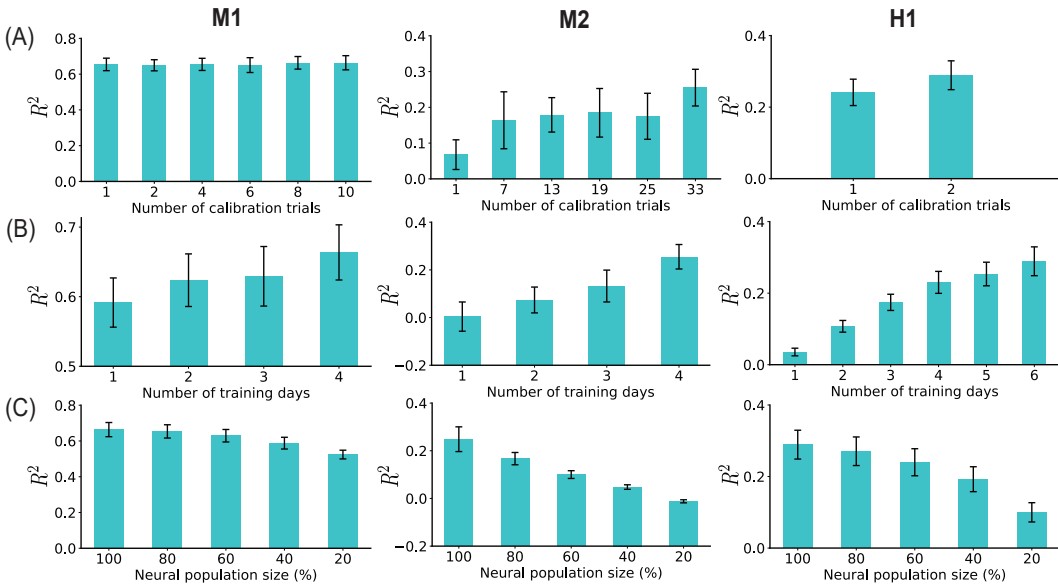

Figure 3: **Scaling analyses**. Cross-session performance of SPINT against number of calibration trials (A), training days (B), and population sizes (C) across M1, M2, and H1 datasets. Bars represent mean $R^2$ across held-out sessions, whiskers represent standard error of the mean of $R^2$ across held-out sessions.

### 4.5 SPINT performance scales well with the amount of training data

Thanks to the flexible permutation-invariant transformer network and the context-dependent positional embeddings, SPINT can ingest populations with arbitrary sizes and orders. These design choices give SPINT the ability to scale naturally with large amounts of training data. We demonstrate this scaling ability in Figure 3(B), where we observe a clear trend in cross-session performance when we use data from more held-in days to train the model, with the best performance achieved when using all available training data on each dataset. This ability suggests potentials of SPINT as a large-scale pretrained model for iBCI when trained on larger datasets beyond the FALCON benchmark.

### 4.6 SPINT is robust to variable population composition

Our proposed dynamic channel dropout encourages robustness of SPINT to variable input population size and membership. We test this robustness by training SPINT on the full held-in populations with dynamic channel dropout and evaluating on variable-sized held-out populations (Figure 3C). At each evaluation batch, we randomly sample a subset of the original population and measure the $R^2$ obtained when the model makes predictions based on this limited subset. We observe robust performance in M1 with reasonable performance drop when the population gets increasingly smaller, with the model still achieving a mean held-out $R^2$ of $0.52$ when only $20\%$ of the original population remains, outperforming other ZS and FSU baselines with the full population.

### 4.7 SPINT maintains low-latency inference for iBCI systems

A critical consideration in iBCI system deployment is the ability of the system to perform behavior decoding in real time. We designed SPINT with this consideration in mind, using only one layer of cross-attention and two three-layer fully connected networks for IDEncoder. In Table 2, we report the latency achieved by SPINT as compared to other methods. Latency is defined as the amount of time a method requires to process the evaluation data divided by the duration of the evaluation data [72]. The ratio less than $1$ signifies the approximation to real-time iBCI inference. SPINT achieves $0.13$ latency on M1 and M2, and $0.14$ latency on H1, matching or outperforming transformer baselines, while being significantly below $1$. In practice, SPINT could be potentially faster in terms of deployment time, as it eliminates the need for an explicit alignment step required by conventional iBCI systems.

| | Class | M1 | M2 | H1 |
|---|---|---|---|---|
| Wiener Filter (WF) | OR | 0.06 | 0.08 | 0.14 |
| RNN | OR | 0.04 | 0.04 | 0.08 |
| NDT2 Multi | OR | 0.15 | 0.10 | 2.29 |
| NDT2 Multi | FSS | 0.13 | 0.10 | 0.30 |
| WF | ZS | 0.06 | 0.08 | 0.15 |
| RNN | ZS | 0.03 | 0.01 | 0.02 |
| CycleGAN + WF | FSU | 0.07 | 0.09 | 0.16 |
| NoMAD + WF | FSU | 0.99 | 0.91 | 1.03 |
| **SPINT (Ours)** | GF-FSU | 0.13 | 0.13 | 0.14 |

Table 2: Inference latency of SPINT against oracles (OR), few-shot supervised (FSS), few-shot unsupervised (FSU) and zero-shot (ZS) methods on held-out sessions (lower is better).

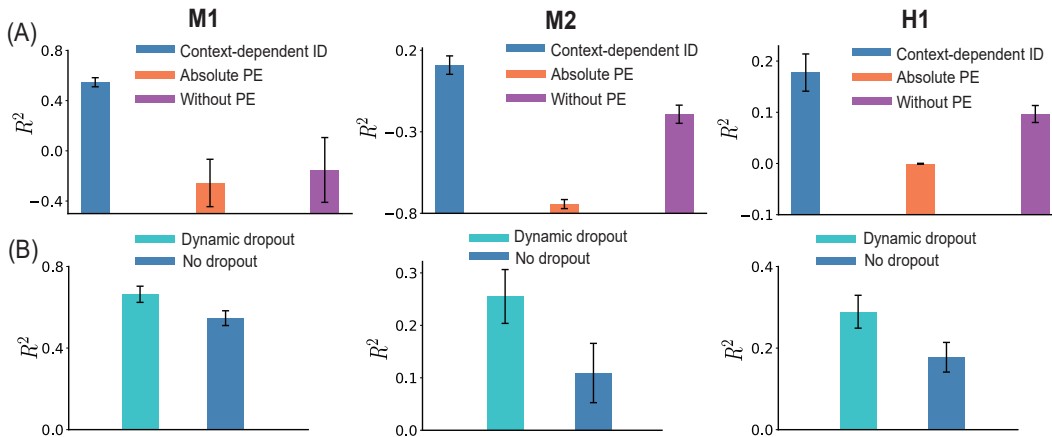

Figure 4: **Ablation Study**. Analyses showing the critical roles of our proposed context-dependent ID against fixed positional embeddings (PE) and no positional embeddings (A), our dynamic channel dropout against no dropout (B). Results are shown across M1, M2, and H1 datasets. Bars represent mean $R^2$ across held-out sessions, whiskers represent standard error of the mean of $R^2$ across held-out sessions.

## 4.8 Ablation Study

We perform ablation studies to demonstrate the benefits of our context-dependent positional embeddings and dynamic channel dropout techniques. In Figure 4A, we compare our context-dependent positional embeddings with fixed (absolute) positional embeddings used in the vanilla transformer [59], and with no positional embeddings. The conventional fixed positional embeddings break the permutation-invariance property of the cross-attention mechanism, thus are not able to generalize to populations with different compositions in held-out sessions. With no positional embeddings, the model is permutation-invariant by design; however, the loss of information about neural unit functional identities hinders the model's ability to decode the behavior these units encode. We achieve the best of both worlds by our proposed context-dependent positional embeddings, being both permutation-invariant while retaining neural identities for behavior decoding.

To demonstrate the effectiveness of our proposed dynamic channel dropout technique, we compare the cross-session performance of SPINT with dynamic channel dropout and without dropout. We show in Figure 4B that dynamic channel dropout serves as an effective regularization technique by preventing the model from overfitting to the population composition in training sessions.

## 5 Discussion

In this work, we introduce SPINT, a permutation-invariant transformer designed for cross-session intracortical motor decoding. SPINT features a context-dependent neural ID embeddings that dynamically infers unit identities, allowing the model to handle unordered, variable-sized populations

across sessions. These components enable SPINT to adapt to new sessions using only a small amount of unlabeled calibration data, with no need for test-time labels or gradient updates. We demonstrate SPINT's cross-session generalization ability on three continuous motor decoding tasks from the FALCON Benchmark, where it consistently outperforms existing zero-shot and few-shot unsupervised baselines, even surpassing few-shot supervised and oracle models in some instances. SPINT's light-weight design also enables low-latency inference, making it well-suited for real-world iBCI applications.

Our study represents an initial attempt at a flexible, gradient-free framework for consistent behavioral decoding and opens up several promising avenues for future research. Within the framework of the FALCON Benchmark, we demonstrated that our approach scales effectively with increasing amount of training data, albeit from a single subject and behavior task. Exploring the applicability of our approach under more diverse settings (cross-task, cross-region, cross-subject) and on non-motor behaviors is an important direction for future research [44, 22, 47, 48, 49]. On the technical side, our end-to-end training for the unit identifier together with the rest of the network ties the identification of the neural identities to task-specific behavioral decoding and thus requires access to labels to train the unit identifier, which might be limited in real-world iBCI settings. Disentangling the training of the unit identifier with behavioral decoding by means of self-supervised approaches such as contrastive or predictive learning can potentially alleviate the reliance on behavior labels for this stage, albeit at the cost of additional training overhead and potential loss of behavior-relevant neural features essential for downstream decoding.

Lastly, although our study shows robust and low-latency behavioral decoding *in silico*, its performance remains to be evaluated *in vivo*, where sensory feedback from closed-loop control may induce modulation in the neural activity inputs and hardware-specific constraints could affect the online inference latency [75]. To this end, our preliminary assessment represents an initial step toward a light-weight, plug-and-play decoding framework for long-term iBCI systems.

# 6 Acknowledgment

The authors acknowledge the partial support of HDR Institute: Accelerated AI Algorithms for Data-Driven Discovery (A3D3) National Science Foundation grant PHY-2117997 (TL, HF, JL, AO, ES) and EFRI-BRAID-2223495 (ES, TL). The authors also acknowledge the partial support by the Departments of Electrical Computer Engineering, Applied Mathematics and Bioengineering. The authors are thankful to the Center of Computational Neuroscience, the eScience Center at the University of Washington and the Allen Institute for Brain Science. TL and US wish to thank the founder of the Allen Institute, P.G. Allen, for his vision, encouragement, and support. The authors also thank the FALCON organizers Joel Ye, Brianna Karpowicz, Chethan Pandarinath, Thiago Scodeler, and the EvalAI team for their technical support with the datasets and submissions.

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

# A  Appendix

## A.1  Within-session performance comparison

We include the within-session performance comparison between SPINT and baselines in Table A1. This table is similar to Table 1 in the main paper, but with metrics obtained on EvalAI's private splits within the held-in sessions. As observed from the table, SPINT also consistently outperforms ZS and FSU baselines on the held-in splits.

|  | **Class** | **M1** | **M2** | **H1** |
|---|---|---|---|---|
| Wiener Filter (WF) | OR | $0.54 \pm 0.01$ | $0.27 \pm 0.02$ | $0.24 \pm 0.02$ |
| RNN | OR | $0.75 \pm 0.03$ | $0.59 \pm 0.07$ | $0.51 \pm 0.09$ |
| NDT2 Multi [21] | OR | $0.77 \pm 0.03$ | $0.62 \pm 0.03$ | $0.68 \pm 0.05$ |
| NDT2 Multi [21] | FSS | $0.77 \pm 0.03$ | $0.63 \pm 0.03$ | $0.62 \pm 0.04$ |
| WF | ZS | $0.46 \pm 0.06$ | $0.15 \pm 0.07$ | $0.20 \pm 0.04$ |
| RNN | ZS | $0.52 \pm 0.15$ | $0.20 \pm 0.29$ | $0.31 \pm 0.13$ |
| CycleGAN + WF [29] | FSU | $0.61 \pm 0.02$ | $0.32 \pm 0.03$ | $0.15 \pm 0.04$ |
| NoMAD + WF [25] | FSU | $0.64 \pm 0.01$ | $0.35 \pm 0.05$ | $0.21 \pm 0.06$ |
| **SPINT (Ours)** | GF-FSU | $\mathbf{0.77} \pm 0.02$ | $\mathbf{0.59} \pm 0.01$ | $\mathbf{0.47} \pm 0.06$ |

Table A1: Within-session performance comparison against oracles (OR), few-shot supervised (FSS), few-shot unsupervised (FSU), and zero-shot (ZS) methods. Our SPINT approach belongs to a special class which we termed Gradient-Free Few-Shot Unsupervised (GF-FSU), where models perform adaptation based on few-shot unlabeled data but *without* any parameter updates at test time. Results are reported as mean $\pm$ standard deviation $R^2$ across held-in sessions, achieved on EvalAI private held-in splits.

## A.2  Proof of SPINT's permutation-invariance

Let $P_R$, $P_C$ be the row and column permutation matrices of the same permutation $\pi$ ($P_C = P_R^\top = P_R^{-1}$ and $P_C P_R = I$). Also let $X' = P_R X$ and $(X^C)' = P_R X^C$ be the row-permuted neural windows and row-permuted calibration trials.

Since the ID embedding of each neural unit $i$ is computed individually from the set of calibration trials for that unit:

$$E_i = \text{IDEncoder}(X_i^C) = \psi(pool(\phi(X_i^C))), \tag{A1}$$

permuting the neural units in the original population (neural windows $X$ or calibration trials $X^C$) will permute the embedding matrix $E$ in the exact same order, i.e., $E' = P_R E$.

It follows that:

$$Z' = X' + E' = P_R X + P_R E = P_R(X + E) = P_R Z \tag{A2}$$

In other words, $Z$ is equivariant to the permutation of neural units.

Cross-attention performed on $Z'$ then becomes:

$$
\begin{aligned}
\text{CrossAttn}(Q, Z', Z') &= \text{CrossAttn}(Q, P_R Z, P_R Z) \\
&= \text{softmax}\left(\frac{QW_K^\top Z^\top P_R^\top}{\sqrt{d_k}}\right) P_R Z W_V \\
&= \text{softmax}\left(\frac{QW_K^\top Z^\top P_C}{\sqrt{d_k}}\right) P_R Z W_V \\
&= \text{softmax}\left(\frac{QW_K^\top Z^\top}{\sqrt{d_k}}\right) P_C P_R Z W_V \\
&= \text{softmax}\left(\frac{QW_K^\top Z^\top}{\sqrt{d_k}}\right) Z W_V \\
&= \text{CrossAttn}(Q, Z, Z)
\end{aligned}
\tag{A3}
$$

where $\text{softmax}\left(\frac{QW_K^\top Z^\top P_C}{\sqrt{d_k}}\right) = \text{softmax}\left(\frac{QW_K^\top Z^\top}{\sqrt{d_k}}\right)P_C$ because an element is always normalized with the same group of elements in the same row regardless of whether column permutation is performed before or after $\text{softmax}$.

Equation A3 concludes Proposition 1 in the main paper.

We note that multi-layer perceptron (MLP), layer normalization, and residual connection are applied row-wise and hence do not affect the overall permutation-invariance property of our SPINT framework.

### A.3 Correlation of attention scores and firing statistics

We ask whether the attention scores SPINT assigns for each neural unit are correlated with its firing statistics. To answer this question, in each held-out calibration window, we measure the average attention scores over $B$ behavior covariates, and its firing statistics (mean/standard deviation) over the held-out calibration trials, then calculate the Pearson's correlation between these two quantities using all held-out calibration windows. We show the results in Table A2.

We observe that the attention scores correlate moderately with the mean and the standard deviation of the neural unit's firing rates, with higher correlation for the standard deviation than the mean, suggesting that SPINT might be extracting neural units that are active (having high mean firing rates) and behaviorally relevant (having high variance throughout the calibration periods where behavior is varied) to pay attention to in behavioral decoding.

|  | M1 | M2 | H1 |
|---|---|---|---|
| $\rho$(attention scores, mean firing rates) | $0.33 \pm 0.16$ | $0.76 \pm 0.03$ | $0.51 \pm 0.04$ |
| $\rho$(attention scores, standard deviation of firing rates) | $0.45 \pm 0.16$ | $0.87 \pm 0.02$ | $0.57 \pm 0.03$ |

Table A2: Pearson's correlation between attention scores for each neural unit and that unit mean/standard deviation of firing rates during the held-out calibration periods. Results are reported as the mean correlation $\pm$ standard deviation across held-out sessions. All $p$-values are less than $0.05$.

### A.4 Implementation details

#### A.4.1 Data preprocessing

For neural activity, we use the binned spike count obtained by unit threshold crossing with the standard bin size of 20ms as set forth by the FALCON Benchmark. We follow FALCON's continuous decoding setup for all three M1, M2, and H1 datasets, where rather than decoding trialized behavior from the trialized neural activity (often performed in a non-causal manner), we decode behavior at the last step of a neural activity window, mimicking the online, causal iBCI decoding. To construct the length-$W$ neural window at the beginning of each session, we pre-pad the session neural time series with $(W-1)$ zeros. We discard the windows whose last time step belongs to a non-evaluated period as defined by FALCON, e.g., inter-trial periods where there is no registered kinematics.

Our IDEncoder infers neural unit identity from trialized calibration trials. As calibration trials vary in length, we interpolate all calibration trials to the same length $T$, where $T = 100$ for M2 and $T = 1024$ for M1 and H1. We use the Python library `scipy.interpolate.interp1d` with a cubic spline for interpolation. Note that we only perform interpolation for neural calibration trials to synchronize their trial lengths. We still use the raw spike counts for the neural windows, conforming with the continuous decoding setup.

#### A.4.2 Behavior output scaling

For M2 and H1, since values of behavior covariates are relatively small, during training we scale the network behavior predictions by a factor of $0.2$ and $0.05$ for M2 and H1, respectively, effectively asking the model to predict $5\times$ and $20\times$ the original behavior values. The MSE loss and $R^2$ metrics are computed between the scaled predicted outputs and the original ground truth values.

| Dropout | 0 | 0.2 | 0.4 | 0.6 | 0.8 | DD [0,1] |
|---|---|---|---|---|---|---|
| $R^2$ | $0.51 \pm 0.13$ | $0.62 \pm 0.10$ | $0.63 \pm 0.10$ | $0.63 \pm 0.10$ | $0.60 \pm 0.09$ | $0.64 \pm 0.10$ |

Table A3: SPINT's cross-session performance against dynamic dropout and different choices of fixed dropout rates. Results are reported as mean $\pm$ standard deviation across held-out calibration sessions. DD [0,1] stands for dynamic dropout with variable dropout rates between 0 and 1.

| DD range | [0, 0.1] | [0, 0.2] | [0, 0.3] | [0, 0.4] | [0, 0.5] | [0,1] |
|---|---|---|---|---|---|---|
| $R^2$ | $0.59 \pm 0.07$ | $0.59 \pm 0.07$ | $0.61 \pm 0.10$ | $0.62 \pm 0.07$ | $0.63 \pm 0.07$ | $0.64 \pm 0.10$ |

Table A4: SPINT's cross-session performance across different ranges of dynamic dropout. Results are reported as mean $\pm$ standard deviation across held-out calibration sessions.

| # heads | 4 | 8 | 16 | 32 | 64 |
|---|---|---|---|---|---|
| $R^2$ | $0.62 \pm 0.08$ | $0.63 \pm 0.09$ | $0.64 \pm 0.10$ | $0.65 \pm 0.11$ | $0.64 \pm 0.10$ |

Table A5: SPINT's cross-session performance for different cross-attention head counts. Results are reported as mean $\pm$ standard deviation across held-out calibration sessions.

| # self-attention layers | 0 | 1 | 2 | 3 | 4 |
|---|---|---|---|---|---|
| $R^2$ | $0.64 \pm 0.10$ | $0.63 \pm 0.13$ | $0.57 \pm 0.13$ | $0.61 \pm 0.10$ | $0.60 \pm 0.15$ |

Table A6: SPINT's cross-session performance for different number of self-attention layers. Results are reported as mean $\pm$ standard deviation across held-out calibration sessions.

| # cross-attention layers | 1 | 2 | 3 | 4 | 5 |
|---|---|---|---|---|---|
| $R^2$ | $0.64 \pm 0.10$ | $0.65 \pm 0.10$ | $0.65 \pm 0.10$ | $0.64 \pm 0.11$ | $0.62 \pm 0.13$ |

Table A7: SPINT's cross-session performance for different number of cross-attention layers. Results are reported as mean $\pm$ standard deviation across held-out calibration sessions.

| Window size | 50 | 100 | 200 | 400 | 600 |
|---|---|---|---|---|---|
| $R^2$ | $0.65 \pm 0.10$ | $0.64 \pm 0.10$ | $0.64 \pm 0.10$ | $0.60 \pm 0.10$ | $0.61 \pm 0.09$ |

Table A8: SPINT's cross-session performance for different context window sizes. Results are reported as mean $\pm$ standard deviation across held-out calibration sessions.

### A.4.3 Inferring neural unit identity

We follow the permutation-invariant framework in [56] for inferring identity $E_i$ of neural unit $i$:

$$E_i = \text{IDEncoder}(X_i^C) = \text{MLP}_2\big(\frac{1}{M}\sum_{j=1}^{M}(\text{MLP}_1(X_i^{C_j}))\big) \tag{A4}$$

where $M$ is the number of calibration trials, $X_i^{C_j}$ is the neural activity of the $j^{th}$ calibration trial of neural unit $i$, $\text{MLP}_1$ and $\text{MLP}_2$ are two 3-layer fully connected networks. $\text{MLP}_1$ projects the length-$T$ trials to a hidden dimension $H$, and $\text{MLP}_2$ projects the length-$H$ hidden features to length-$W$ neural unit identity output.

### A.4.4 Behavioral decoding by cross-attention

After neural identity for all units $E$ is inferred, we add it to the neural window input $X$ to form the identity-aware neural activity $Z$, i.e., $Z = X + E$. We then use the cross-attention mechanism in the latent space to decode last step behavior covariates. Specifically:

$$Z_{in} = \text{MLP}_{in}(Z) \tag{A5}$$

$$\tilde{Z} = Q + \text{CrossAttn}(Q, \text{LayerNorm}(Z_{in}), \text{LayerNorm}(Z_{in})) \tag{A6}$$

$$Z_{out} = \tilde{Z} + \text{MLP}_{attn}(\text{LayerNorm}(\tilde{Z})) \tag{A7}$$

$$Y = \text{MLP}_{out}(Z_{out}) \tag{A8}$$

### A.4.5 Hyperparameters

We include the notable hyperparameters used to optimize SPINT in Table A9. We train and evaluate models for each M1, M2, and H1 dataset separately. We train the models using all available held-in sessions and evaluate on all available held-out sessions. We use Adam optimizer [76] for all training.

|                                  | M1    | M2    | H1    |
|----------------------------------|-------|-------|-------|
| Batch size                       | 32    | 32    | 32    |
| Window size                      | 100   | 50    | 700   |
| Max trial length                 | 1024  | 100   | 1024  |
| Number of IDEncoder layers       | 3, 3  | 3, 3  | 3, 3  |
| Number of cross attention layers | 1     | 1     | 1     |
| Hidden dimension                 | 1024  | 512   | 1024  |
| Behavior scaling factor          | 1     | 0.2   | 0.05  |
| Learning rate                    | 1e−5  | 5e−5  | 1e−5  |

Table A9: Hyperparameters used to train SPINT on the M1, M2, and H1 datasets.

We include representative hyperparameter sweeps demonstrating SPINT's robustness to hyperparameter choices in Tables A3, A4, A5, A6, A7, A8. This robustness allows SPINT to effectively capture long-range context while maintaining a minimalist architecture without compromising generalizability. All sweep results were obtained on 20% of calibration trials held out from each session of the M1 dataset rather than on the EvalAI test split.

### A.4.6 Computational resources

SPINT was trained using a single A40 GPU, consuming less than 2GB of GPU memory with batch size of 32 and taking around 12 hours, 5 hours, and 8 hours to finish 50 training epochs for M1, M2, and H1, respectively. We select checkpoints for evaluation at epoch 50 in all M1, M2, and H1 datasets.

