# OpenReview forum: "SPINT: Spatial Permutation-Invariant Neural Transformer for Consistent Intracortical Motor Decoding"
_NeurIPS.cc/2025/Conference — NeurIPS 2025 poster_

### Official Review · Reviewer_Fnbe · 2025-06-05

**Clarity:** 4
**Significance:** 3
**Originality:** 3
**Rating:** 5
**Confidence:** 4

**Summary:**

Authors introduce SPINT, a neural decoder that is invariant to unit activity, thanks to a context dependent positional encoding. They show improved performance compared to other zero or few-shot methods on the FALCON dataset.

**Questions:**

First, i find section 3.5 a bit unclear. From the rest of the paper and the architecture design, I would have expected that IDEncoder is fine-tuned, while the cross attention layer remains fixed in the few shot experiment. Lines 190-192 seems to indicate the contrary? I am not sure I understand this part correctly and believe this could be clarified further.

Also, authors claim that this is an attempt at a gradient-free framework. I believe that their method could be applicable in more complex datasets as a few-shot decoder (i.e. not gradient free), but don't believe that it will work as a zero-shot method. I would appreciate if authors can explain why they claim their method to be a step towards a gradient-free method, or remove this claim (I don't believe it's a drawback of the paper since few-shot is already sufficient in my opinion, but I would appreciate a further explanation if the method is truly zero-shot). To clarify this further, in Table 1, it would help to show which methods are OR, FSS, FSU, and ZS (you could have four "rows" for each of them and add it in the table).

In Figure 3A, it is very surprising that the method works on M1 with 1 calibration trial only. Could the authors give some intuition behind this result?

The authors claim that the neural population can be of arbitrary size. Isn't it bounded by a maximum population size? How do you deal with populations of increasing size? The problem of having "disappearing neurons" is adressed, but neurons can appear in the recordings as well (it can also be an artefact of the spike sorting / preprocessing pipeline).

Finally, in in-vivo recordings, often neurons get "mixed up" together, rather than "permuted", in the sense of spikes from one neuron get assigned to the other and vice versa, as a result of the spike sorting pipeline. Do you think the method will be robust to such noise, and could you augment your permutations (maybe add such a perturbation on top of the channel dropout) to make it robust to these artefacts?

**Ethical Concerns:**

["NO or VERY MINOR ethics concerns only"]

**Final Justification:**

I wish to keep my positive score.

**Limitations:**

Yes, the authors have discussed the limitations.
As mentioned, two important challenges are to align across task, regions and subjects, and to extend the method to in vivo recordings.

**Paper Formatting Concerns:**

No formatting concern

**Quality:**

4

**Strengths And Weaknesses:**

Strengths: It is a very good paper, building on a great idea (being invariant to unit ID), with sound design choices and good evaluation.

Weaknesses: I don't see no major weakness to this paper, but as the authors mentioned in the Discussion section, it will be important for future work to evaluate SPINT in vivo, where the noise and perturbations to unit assignment and spike detection can be much more complex.

---

> ### Author Rebuttal · Authors · 2025-07-31
>
> We thank the reviewer for the insightful review, valuable suggestions and constructive feedbacks. We provide clarification to the reviewer questions in our point-by-point responses below:
>
> **Q1. re: clarification on whether IDEncoder is fine-tuned in the new sessions.**
>
> We thank the reviewer for the questions. The IDEncoder is trained with behavior labels during held-in sessions and its weights are frozen during inference on held-out sessions. IDEncoder’s role is to identify units from their temporal activations in calibration trials. We assume this identity is largely preserved for each individual unit across sessions, only their order is changed due to shifts in electrodes between sessions. Therefore, with sufficient training during held-in sessions, IDEncoder can infer this preserved identity out-of-the-box on the held-out sessions without any fine-tuning.
>
> **Q2. re: clarification on gradient-free, zero-shot, and few-shot terminologies**
>
> We thank the reviewer for the comments. By SPINT being a few-shot unsupervised method we mean SPINT would need a few unlabeled calibration trials from the held-out sessions for adaptation. This adaptation does not require any parameter updates in either the IDEncoder or the rest of the model as we elaborate in the response above, therefore SPINT is a *gradient-free* framework. The reviewer is correct that SPINT is not a zero-shot method, as it would still need a few calibration trials at test time. However, unlike existing few-shot baselines where model weights are adapted with gradients at test time (NoMAD and CycleGAN), SPINT can be adapted in the forward-only manner. Therefore we created a new class of method called *gradient-free few-shot unsupervised* to properly distinguish SPINT with the existing baselines.
>
> One model can belong to different model categories (ZS, FSU, FSS, OR), depending on whether they have access to the held-out calibration neural activity and behavior labels. In Table 1 of our paper we followed the reporting format in the FALCON paper [1] to mention each model’s class in the column next to their names.
>
> **Q3: re: intuition how SPINT can work with only one calibration trial on M1**
>
> We agree with the reviewer that this is a surprising and interesting result. The ability to generalize even with one calibration trial can be attributed to the consistency of the animal behavior and the relatively large size of the M1 dataset (5 and 6 times larger than the H1 and M2 datasets, respectively). After being sufficiently trained on a sufficient amount of structured data, when presented with even a single trial at test time, the IDEncoder can extract distinct features from the temporal responses of the units, and the cross-attention mechanism can selectively aggregate these information for behavior decoding.
>
> **Q4: re: robustness of SPINT against test populations with larger size than the training population size**
>
> We appreciate the reviewer’s thoughtful question. We conducted additional experiments on to gauge the robustness of SPINT when the population is larger than the size seen during training. Specifically, we artificially duplicated a random subset of units in the population and evaluated SPINT’s cross-session performance on this larger population. We found that SPINT is robust to the increased population size, even up to twice the size of original population (Table 1). Since the EvalAI platform was down during the rebuttal period, we show here the performance on the public calibration portion of the held-out sessions on M1 dataset.
>
> | Population size | 100%           | 120%            | 140%           | 160%           | 180%           | 200%           |
> |-----------------|----------------|-----------------|----------------|----------------|----------------|----------------|
> | Held-out $R^2$  | 0.64$\pm$ 0.10 | 0.63 $\pm$ 0.09 | 0.63$\pm$ 0.09 | 0.63$\pm$ 0.09 | 0.63$\pm$ 0.08 | 0.63$\pm$ 0.09 |
>
> Table 1: SPINT robustness to increased size of population. Results are reported as mean $\pm$ standard deviation across held-out calibration sessions.
>
> **Q5. re: robustness of SPINT against noise in spike assignment**
>
> We thank the reviewer for raising an interesting question. We agree with the reviewer that noise introduced by the spike sorting pipeline could negatively impact SPINT performance. We conducted additional experiments on the M1 dataset where we artificially shuffle spikes among a subset of random units and evaluated SPINT cross-session performance:
>
> | Augmentation rate | 0.0           | 0.05          | 0.1            | 0.2           | 0.4           |
> |-------------------|---------------|---------------|----------------|---------------|---------------|
> | Held-out $R^2$    | 0.64$\pm$0.10 | 0.61$\pm$0.08 | 0.57$\pm$ 0.09 | 0.53$\pm$0.08 | 0.38$\pm$0.05|
>
> Table 2: SPINT robustness to noise in spike assignment. Results are reported as mean $\pm$ standard deviation across held-out calibration sessions.
>
> This is somewhat expected, as SPINT relies on consistent spiking features in a unit’s responses to accurately identify the unit. With spikes being assigned to incorrect units, this ability to extract unit identity would be impacted, leading to a degraded decoding performance as a result. Our initial attempts to introduce this noise corruption during training did not result in improved performance; however, we agree with the reviewer that this is an interesting avenue to explore in future extensions of the method.
>
>
> [1] Brianna Karpowicz, Joel Ye, Chaofei Fan, Pablo Tostado-Marcos, Fabio Rizzoglio, Clayton Washington, Thiago Scodeler, Diogo de Lucena, Samuel Nason-Tomaszewski, Matthew Mender, et al. Few-shot algorithms for consistent neural decoding (falcon) benchmark. Advances in Neural Information Processing Systems, 37:76578–76615, 2024.

---

> ### Comment · Reviewer_Fnbe · 2025-08-01
>
> I thank the reviewers for answering my questions.
>
> I had only minor comments and did not see any major weaknesses when reviewing the paper.
>
> I understand Reviewer MJ6k's response, but believe that the task of "cross-session nonstationarity within a subject" is a very important task. However, when comparing to other methods, I agree that looking at cross-subject generalization is very important as competing methods may be designed for this task, and it's important to properly state the conceptual differences with POYO, NDT-3 and other competing tasks. I would recommend making this crystal clear in the manuscript.
>
> For now, I wish to keep my score and recommend acceptance.

---

> > ### Author Response · Authors · 2025-08-03
> >
> > We thank the reviewer for taking time to review our paper and rebuttal, and for acknowledging the significance of our work with acceptance recommendation! We appreciate the reviewer's constructive feedbacks which greatly helped improve our manuscript, and will follow the reviewer's suggestions to state the conceptual differences between SPINT and other models in the revised version of our paper :)

---

### Official Review · Reviewer_CKq1 · 2025-07-02

**Clarity:** 3
**Significance:** 2
**Originality:** 3
**Rating:** 4
**Confidence:** 1

**Summary:**

The paper tackles cross‐session drift in intracortical BCIs caused by shifting neural signals. It introduces SPINT, a transformer that treats neurons as unordered tokens and infers their identities from a small set of unlabeled calibration data. SPINT uses context‐dependent position embeddings and dynamic channel dropout to adapt without gradient updates. It outperforms zero‐ and few‐shot baselines on multi‐session motor decoding benchmarks.

**Questions:**

1. How well does SPINT handle very large shifts in neural signals between sessions?
2. Can SPINT adapt with only one or two calibration trials?
3. Could this method work for non-motor tasks (e.g., speech decoding)?
4. Are there any privacy or ethical concerns with long-term neural recording?

I will take into account other reviewers’ opinions when evaluating the paper further.

**Ethical Concerns:**

["NO or VERY MINOR ethics concerns only"]

**Final Justification:**

After reading other reviewers' reviews and authors' rebuttal. I agree that this work is acceptable.

**Limitations:**

No.

The authors do outline methodological limitations in the Discussion section, but they omit any consideration of potential negative or dual-use societal impacts.

Suggestions for improvement:
1. Add a “Broader Impacts” subsection that discusses risks of misuse (e.g., privacy breaches, uneven access to iBCI technologies).
2. Reflect on ethical concerns around long-term neural recording and consent, and outline safeguards (e.g., data anonymization, IRB oversight).
3. Consider potential downstream harms (e.g., neuro-surveillance) and propose mitigation strategies.

**Quality:**

2

**Strengths And Weaknesses:**

**Strengths**
1. The paper is well written and organized, with diagrams and pseudo-code that make the approach easy to follow.
2. The experiments are comprehensive and rigorously compare SPINT to zero-shot, few-shot, and oracle baselines on three primate and human datasets, showing clear performance gains.
3. SPINT adapts with only a small amount of unlabeled calibration data and requires no gradient updates, making it practical for real-world BCIs.
4. Combining context-dependent position embeddings with dynamic channel dropout is a novel solution for session drift without explicit alignment.

**Weaknesses**
1. No failure cases are shown, so it’s unclear how SPINT handles extreme signal drift or very limited calibration data.
2. Hyperparameter choices—such as dropout rates and attention head counts—are not well justified, making sensitivity hard to assess.
3. Evaluation is limited to motor decoding, leaving its generality to other BCI tasks untested.

---

> ### Author Rebuttal · Authors · 2025-07-31
>
> We thank the reviewer for providing valuable suggestions and feedbacks on our manuscript. We address the reviewer's concerns and questions in our point-by-point responses below:
>
> **W1. re: failure cases**
>
> We thank the reviewer for the comments. We show in the paper that SPINT can handle well the signal drift across held-in and held-out sessions, which can be three weeks to one month apart in the extreme cases of all three datasets. We also showed in Figure 3 our analyses on how SPINT performed with very limited calibration data. Except for the M1 dataset where SPINT can perform well even with one calibration trial, we observed that with one calibration trial in M2 and H1, SPINT performance degraded with held-out $R^2$ around zero in these extreme cases.
>
> **W2. re: sensitivity to hyperparameter choices**
>
> We appreciate the reviewer’s request for more details on the hyperparameter choices of fixed dropout rates and attention head counts. While we employed dynamic dropout instead of the conventional fixed dropout, we provide below the sweeping analysis on different choices for fixed dropout rate together with different range of dynamic dropout rates for reference. We also include sweeping experiments on attention head counts as the reviewer suggested. SPINT’s performance on held-out sessions remains stable across different choices of dropout rates and attention head counts. The results demonstrate SPINT’s robustness to the choices of hyperparameters.
>
> | Fixed dropout rate | 0             | 0.2           | 0.4           | 0.6           | 0.8           | dynamic dropout [0, 1] |
> |---------------|---------------|---------------|---------------|---------------|---------------|------------------------|
> | $R^2$         | 0.51$\pm$0.13 | 0.62$\pm$0.10 | 0.63$\pm$0.10 | 0.63$\pm$0.10 | 0.60$\pm$0.09 | 0.64$\pm$0.10          |
>
> Table 1: SPINT's cross-session performance against dynamic dropout and different choices of fixed dropout rates. Results are reported as mean $\pm$ standard deviation across held-out calibration sessions.
>
> | Dynamic dropout ranges | [0, 0.1]             | [0, 0.2]           | [0, 0.3]          | [0, 0.4]          | [0, 0.5]           | [0, 1] (Ours)|
> |---------------|---------------|---------------|---------------|---------------|---------------|------------------------|
> | $R^2$         | 0.59$\pm$0.07 | 0.59$\pm$0.07 | 0.61$\pm$0.10 | 0.62$\pm$0.07 | 0.63$\pm$0.07 | 0.64$\pm$0.10          |
>
> Table 2: SPINT's cross-session performance against different choices of range for dynamic dropout. Results are reported as mean $\pm$ standard deviation across held-out calibration sessions.
>
> | # Attention heads | 4             | 8             | 16            | 32            | 64            |
> |-------------------|---------------|---------------|---------------|---------------|---------------|
> | $R^2$             | 0.62$\pm$0.08 | 0.63$\pm$0.09 | 0.64$\pm$0.10 | 0.65$\pm$0.11 | 0.64$\pm$0.10 |
>
> Table 3: SPINT's cross-session performance against different choices of attention head counts. Results are reported as mean $\pm$ standard deviation across held-out calibration sessions.
>
> **W3. re: evaluation limited to motor decoding**
>
> We appreciate the reviewer’s concern. While this is driven mainly by the availability of standardized datasets and evaluation platforms, we will mention this perspective in the Limitations section of the revised manuscript.
>
> **Q1. re: SPINT handling of large shift between sessions**
>
> Please see our response in W1
>
> **Q2. re: whether SPINT can adapt with one or two trials**
>
> Yes SPINT can adapt with as few as one or two calibration trials. We showed in Figure 3 of our main paper SPINT’s performance on M1 and H1 datasets when the number of calibration trials are restricted to these values.
>
> **Q3. re: whether SPINT can work on non-motor tasks**
>
> In the absence of actual experiments, we speculatively think that SPINT’s lightweight and neuronal identity based approach will be useful in non-motor tasks as well.
> Our algorithm can be extended to speech and text coding with the following adjustments. In general, for speech/text coding, the data is of variable length and has sparse labels. To address the above difficulties, we can 1) apply zero-padding or learnable padding for the variable-length data so they can have the same maximum length. This padding batch trick is widely used in LLM that can be easily incorporated in our data processing pipelines. 2) The issue of no dense label in text/speech decoding can be solved via the connectionist temporal classification (CTC) loss during model training. In the model inference phase, one can use either greedy or beam search approaches for the final character/phoneme decoding. This will serve as an exciting future extension of our work.
>
> **Q4. re: privacy and ethical concerns of long-term neural decoding**
>
> We work on public datasets within the framework of the FALCON Benchmark and focused more on algorithm development to realize the goal of consistent motor decoding. Thus, our in-silico experiments did not involve any human or animal experiments. In terms of data privacy and ethics protocol in the collection of neural data recording, the FALCON Benchmark has properly addressed these points in their publication.
>
> **L1. re: broader impacts**
>
> We thank the reviewer for suggesting this point. We added them in the discussion section by considering 1) the ethical concerns in iBCI data collection; 2) privacy of iBCI data usage and distribution; 3) the safety of iBCI technology.
>
> Broader Impacts. iBCI technologies raise significant ethical and societal considerations that require proactive mitigation strategies. First, we advocate for consistent consent protocols and enhanced IRB oversight to address challenges of long-term neural data recording and collection, particularly for vulnerable populations with potential neurological and neuropsychiatric disorders. Second, neural data's inherently identifying nature necessitates privacy-preserving techniques beyond traditional anonymization, including different privacy regularizations to prevent risky inference of subject information. Third, we acknowledge dual-use risks including potential neuro-surveillance applications and cognitive manipulation, alongside equity concerns where high costs may exacerbate healthcare disparities. To mitigate these risks, we recommend developing open-source frameworks, establishing international governance standards similar to those proposed for AI-related algorithms and systems, implementing technical safeguards against unauthorized access, and conducting regular ethical impact assessments. To this end, we hope the above considerations can underscore the need for interdisciplinary collaboration between technologists, ethicists, and policymakers to ensure responsible development and deployment of future iBCI technologies.

---

### Official Review · Reviewer_MJ6k · 2025-07-03

**Clarity:** 3
**Significance:** 2
**Originality:** 2
**Rating:** 4
**Confidence:** 4

**Summary:**

This paper introduces SPINT, a spatial permutation-invariant transformer architecture for behavioural decoding from neural population activity, in order to tackle the problem of dynamically inferring neural unit identities on new sessions or days and adapting to non-stationarities over time in intracortical brain-computer interface applications. SPINT uses a neural unit identity encoder that is equivariant under permutation of neural unit/channel orderings in the dataset/recording source, and invariant to the ordering of the calibration trials. While this encoder is trained on the pretraining dataset, it is not finetuned when generalising to subsequent sessions or days and is thus used in a zero-shot parameter update-free manner (only the unit identity embeddings are updated using the calibration trials), which the authors refer to as their few-shot unsupervised and gradient-free unit identification scheme. The authors also use a dynamic dropout scheme to improve robustness to variable neural population sizes. Post identity encoding of a time chunk of neural activity, behaviour is decoded using a cross-attention readout. The authors demonstrate results on 3 motor decoding datasets of the FALCON Benchmark for spiking activity-based neural decoders (2 nonhuman primate and 1 human dataset), and demonstrate better performance compared to other unsupervised baseline methods, while also showing some improvements with increasing training data and calibration trials, and with comparable latency to some Transformer-based methods.

**Questions:**

* Could the authors clarify details on the dropout ablation?
* What if you were to train or finetune the `IDEncoder` MLPs with the calibration data since labels are available for those trials? Does it max out performance/lead to considerable gains?
* The model has no self-attention or other intermediate layers, the embedded units seem to be directly used to decode behavior by a fairly small cross-attention layer. Were additional parameters not helpful or necessary to see better performance? Did you try scaling up the model size? I anticipate that this may not help much in the single dataset/subject setting, but in the large-scale multi-dataset/subject pretraining case, I anticipate that smaller models will underperform larger overparameterised models.
* Details on the learnable behaviour queries seem sparse. Could the authors specify additional details on the behaviour queries? Given that only one behaviour at the final timestamp is decoded per context window of neural activity, is there only one behaviour query (initialised at random and learnt during pretraining)? Could there be multiple queries for multiple task readouts?
* The approach currently uses binned spike counts but especially given the cross-attention's ability to handle variable number of input tokens, binning isn't necessary and the authors could use individual spike tokenisation à la POYO. Have the authors tried this scheme? It could potentially yield slightly better performance since there is no loss of information from the binning component, and spike timings could be important [15].
* I see from Table S3 that quite a bit of context is needed for some tasks (M1: $W$ = 100 = 2s, M2: $W$ = 50 = 1s – which is fine, i.e., on par with NDT-2 and POYO, H1: $W$ = 700 = 14s – a lot!), I understand this can work in a sliding window but I'm just curious. Have the authors tried varying these values/doing a search for the best values? Why is so much context needed for some of these tasks?

15. Xie, Weizhen, et al. "Neuronal sequences in population bursts encode information in human cortex." Nature (2024): 1-8.

**Ethical Concerns:**

["NO or VERY MINOR ethics concerns only"]

**Final Justification:**

The authors addressed my questions, clarified their motivation and goals, and performed several experiments during the rebuttal phase. I think the method is neat and shows some good results on FALCON, a standard benchmark, so I'm happy to recommend acceptance.

**Limitations:**

Yes, the authors have discussed the limitations of their work. There is no discussion on negative societal impact. While powerful iBCI decoders may generally have a positive impact on society, it may be worth recognising that a lot of these advances have been born out of animal research and may be misused as well; in addition there are concerns about the privacy of what is being decoded and when e.g. in decoding speech in humans.

**Paper Formatting Concerns:**

No concerns.

**Quality:**

2

**Strengths And Weaknesses:**

**Strengths**

* The writing and figures are clear and easy to understand.
* The authors have performed experiments on a recently introduced standard benchmark for neural decoding and have included important ablations and few-shot + data scaling results.
* The method works with variable-sized neural populations thanks to the cross-attention mechanism.
* The method is simple and can be used for lightweight, update-free adaptation – a key asset – across days in single subject and single task setups.

**Weaknesses**

* All experimental results are on single subject evaluation setups, with no experiments on cross-subject transfer for heldout subjects. While this is a consequence of the FALCON benchmark not having multi-subject data (all of M1, M2, and H1 are single subject datasets), it would be important to study how well SPINT performs in the multi-subject setting using other datasets as done in prior work [1-3] without which it is hard to evaluate the claims of it being a "plug and play" decoding method. The lack of this evaluation also diminishes the significance of the few-shot finetuning paradigm: the model cannot take advantage of multi-dataset pretraining and would have to be retrained for every new subject from scratch, limiting the finetuning scheme to the single subject setting across sessions/days. Baselines discussed in the paper, such as NDT-2 [1], and other key methods such as POYO [2] and NDT-3 [3] (latter is concurrent work) have studied this multi-subject setting, and some of these works demonstrate strong zero-shot or few-shot performance on new days too [1]. Furthermore, in my opinion, the "plug and play" argument for SPINT vanishes because quite a good amount of pretraining data is needed for good performance in the single subject case (ref. Fig. 3B, especially middle and right panels where multiple days of data are needed to get decent decoding performance), and there is no amortization of these costs through pre-training.
* On a related note, while there are three datasets considered here, all are from a single benchmark while several of the mentioned baselines including the NDT models [1,3] have been tested on other public datasets [4-6]. Evaluating SPINT on these datasets would allow for better comparison to a wider range of existing approaches and improve the evaluations in this paper.
* Another key point to me is the lack of comparison to POYO/POYO+ [2,7] (although the former is cited), given several architectural similarities. POYO can also accept unordered neural units as inputs, where it would learn a channel/unit embedding for each neuron, although through supervised learning for each new dataset. It is thus equivariant to the ordering of units. While SPINT takes in spike counts, where there is one token per unit, POYO is generally provided individual spike tokens, although it may also be used with spike counts [2,7,8] ([8] is concurrent work). While SPINT doesn't use self-attention layers, POYO uses a cross-attention encoder and decoder, and the setup in SPINT is very similar to POYO's behaviour queries and the overall Perceiver IO [9] framework. Note also that POYO uses a unit dropout scheme to improve robustness to population variability, so that is yet another similarity. Given these similarities, and the public availability of POYO code through `torch_brain` (along with a checkpoint) [10], I think it would be important for authors to consider this as a baseline on other public datasets (if not the FALCON benchmark, which POYO has not been submitted to). I do acknowledge that each approach has its own assets and pitfalls: POYO _requires_ supervised finetuning but excels at cross-task and cross-subject transfer, SPINT doesn't require parameter finetuning at all but has not been evaluated in a multi-subject setting.
* The proposed approach which considers fixed $W$-length windows of neural activity may not work for attempted speech decoding in the FALCON H2 dataset, since decoding is from a fixed context window and there is no history as you would have in a recurrent architecture (unlike concurrent work [8]). This dataset only contains trial-level sentence annotations and not dense labels, so chunking up trials (which can be 10s+ long) and decoding phonemes from chunks would probably not yield good performance (as such, H2 only has recurrent baselines on the FALCON leaderboard).
* The unit embedding scheme used here is tied to a single, specific behavioural task on which the model is trained. This limitation is acknowledged in the discussion, but it is a key obstacle to building cross-task models (currently, models would have to be retrained for each task). The authors could consider a readout scheme like POYO+ [7] to allow for multi-task decoding, but the advantages are limited without large-scale multi-dataset pretrainability. I do agree with the authors' suggestion that SSL approaches are the way forward for this problem.
* This is relatively minor compared to my other concerns, but the channel dropout scheme introduced here with variable dropout rates is not entirely novel, and I'm curious about the exact setup used in the ablation study shown in Fig. 4B. To begin with, standard dropout [12] with a fixed dropout rate $p$ would itself dropout a variable number of units in each forward pass, just with a mean fraction of $p$ units dropped out. Secondly, models like POYO [2] and MYOW [11] have used unit dropout schemes with a minimum threshold for the number of units already. Thirdly, adaptive and variable dropout schemes are already prevalent in the deep learning literature [13,14], but have not been as widely adopted as standard dropout, potentially due to limited gains at large scale. Finally, about Fig. 4B, when the authors mention "without" dynamic dropout, do they mean there is no dropout at all, or just standard, fixed $p$ dropout? Of course I would expect consistent gains with (any type of) dropout as opposed to not having dropout at all, but given that the authors list the "dynamic" dropout as a contribution, I would expect an ablation to show performance comparisons with standard unit dropout [2,11] ideally with different values of $p$ in the ablation study. Could the authors clarify this?

**References**

1. Ye, Joel, et al. "Neural data transformer 2: multi-context pretraining for neural spiking activity." Advances in Neural Information Processing Systems 36 (2023): 80352-80374.
2. Azabou, Mehdi, et al. "A unified, scalable framework for neural population decoding." Advances in Neural Information Processing Systems 36 (2023): 44937-44956.
3. Ye, Joel, et al. "A Generalist Intracortical Motor Decoder." bioRxiv (2025): 2025-02.
4. Perich, Matthew G., Juan A. Gallego, and Lee E. Miller. "A neural population mechanism for rapid learning." Neuron 100.4 (2018): 964-976. (https://dandiarchive.org/dandiset/000688?search=000688&pos=1)
5. O’Doherty, Joseph E., et al. "Nonhuman primate reaching with multichannel sensorimotor cortex electrophysiology." Zenodo (2017). (https://zenodo.org/records/3854034)
6. Churchland, Mark M., et al. "Neural population dynamics during reaching." Nature 487.7405 (2012): 51-56. (https://dandiarchive.org/dandiset/000070?search=churchland&pos=2)
7. Azabou, Mehdi, et al. "Multi-session, multi-task neural decoding from distinct cell-types and brain regions." The Thirteenth International Conference on Learning Representations.
8. Ryoo, Avery Hee-Woon, et al. "Generalizable, real-time neural decoding with hybrid state-space models." arXiv preprint arXiv:2506.05320 (2025).
9. Jaegle, Andrew, et al. "Perceiver io: A general architecture for structured inputs & outputs." arXiv preprint arXiv:2107.14795 (2021).
10. Code: https://github.com/neuro-galaxy/torch_brain/tree/main/examples/poyo; checkpoint: https://github.com/neuro-galaxy/poyo
11. Azabou, Mehdi, et al. "Mine your own view: A self-supervised approach for learning representations of neural activity."
12. Srivastava, Nitish, et al. "Dropout: a simple way to prevent neural networks from overfitting." The journal of machine learning research 15.1 (2014): 1929-1958.
13. Ba, Jimmy, and Brendan Frey. "Adaptive dropout for training deep neural networks." Advances in neural information processing systems 26 (2013).
14. Li, Zhe, Boqing Gong, and Tianbao Yang. "Improved dropout for shallow and deep learning." Advances in neural information processing systems 29 (2016).

---

> ### Author Rebuttal · Authors · 2025-07-31
>
> We thank the reviewer for an insightful review, thoughtful questions, and valuable feedbacks. We address the reviewer's concerns and questions below:
>
> **W1:**
> We thank the reviewer for raising the question about SPINT’s ability to transfer across subjects. Our focus in this paper was to isolate the challenge of cross-session nonstationarity within a subject, which remains a central issue in iBCI systems. When we said SPINT contributed an initial step toward a plug-and-play decoding framework for long-term iBCI, we referred specifically to the out-of-the-box cross-session adaptation within a subject, where we demonstrated gradient-free generalization without test-time labels.
> While we designed SPINT specifically for this practical usage, we agree with the reviewer that cross-subject evaluation would bring further values to the method. We therefore conducted additional experiments on a dataset of an extra monkey performing the M1 task (M1-B) (https://dandiarchive.org/dandiset/001209). While this extra dataset is not part of the official FALCON Benchmark and does not have a private held-out test set on EvalAI, the subject in this dataset (Monkey X) performs the same reach-to-grasp task as Monkey L in the official M1-A dataset, and the FALCON organizers curated its data into held-in and held-out sessions following the same standardized format of the benchmark. We did not observe straight-out transferability when evaluating one subject’s checkpoint on another subject’s held-out sessions. However, with full model weight supervised fine-tuning on the new subject’s held-in sessions, the pretrained SPINT quickly adapts to the new subject, maintaining a gap over the model trained from scratch on the new subject in terms of cross-session generalization (Tables 1 & 2). The EvalAI platform was down during the rebuttal period, therefore we show the performance on the public calibration portion of the held-out sessions.
>
> | | Epoch 1 | Epoch 2 | Epoch 3 | Epoch 4 | Epoch 5 | Epoch 50 |
> |-|-|-|-|-|-|-|
> | Monkey X from scratch | 0.59 $\pm$ 0.06 | 0.60$\pm$ 0.07 | 0.60$\pm$0.07 | 0.59$\pm$0.07 | 0.59$\pm$0.07 | 0.64 $\pm$ 0.08 |
> | Monkey L -> Monkey X | 0.62$\pm$0.06   | 0.64$\pm$0.06  | 0.62$\pm$0.07 | 0.63$\pm$0.07 | 0.65$\pm$0.07 | 0.67 $\pm$ 0.08 |
>
> Table 1: Cross-session performance of SPINT on subject X when finetuned from Monkey L vs. trained from scratch. Results are reported as mean $\pm$ standard deviation across held-out calibration sessions.
>
> | | Epoch 1 | Epoch 2 | Epoch 3 | Epoch 4 | Epoch 5 | Epoch 50 |
> |-|-|-|-|-|-|-|
> | Monkey L from scratch | 0.48$\pm$0.10 | 0.52$\pm$0.08 | 0.53$\pm$0.07 | 0.54$\pm$0.08 | 0.56$\pm$0.08 | 0.64 $\pm$ 0.10|
> | Monkey X -> Monkey L | 0.57$\pm$0.07 | 0.56$\pm$0.10 | 0.60$\pm$0.07 | 0.60$\pm$0.07 | 0.60$\pm$0.07 | 0.63 $\pm$ 0.09 |
>
> Table 2: Cross-session performance of SPINT on subject L when finetuned from Monkey X vs. trained from scratch.
>
> These results highlight the subject transferability of SPINT, where cross-session generalization ability on a new subject is maintained and efficiently achieved within a few epochs of fine-tuning.
>
> **W2 & W3:** We chose the FALCON Benchmark for our analyses because it contains standardized splits with clear constraints on the number of test-time samples, label availability, and latency requirements in real-world iBCI systems that we aspire for SPINT. We also appreciate the reviewer highlighting the similarities and differences between SPINT and POYO. We hope the appreciation we have for POYO and other baselines was clear in our paper. To provide further clarity, we will include a discussion on the unique aspects of POYO and SPINT, and will cite POYO+ and POSSM papers in the final version of our paper.
>
> Direct comparison of SPINT and POYO on the FALCON Benchmark is not trivial, as the benchmark’s continuous decoding evaluation paradigm strictly requires participating models to operate on binned spikes, which POYO does not natively support with its unit tokenization scheme. Following the reviewer’s recommendation, we conducted extra experiments to evaluate SPINT on the suggested Perich et al., 2018 dataset [1] that POYO was also trained on. Due to resource constraints and the lack of standardized few-shot evaluation splits on this dataset as we have in FALCON, we selected 10 and 2 sessions of Monkey C performing the Center-Out (CO) reaching task as held-in and held-out sessions, respectively. We reserved 32 trials on each held-out session as the few-shot calibration set. We binned the spike trains with a 20-ms bin width and followed FALCON’s continuous decoding paradigm on this dataset. SPINT demonstrated good cross-session generalizability on this dataset, achieving $R^2$ of 0.83 $\pm$ 0.01 on the heldout sessions, without fine-tuning or using labels in the held-out sessions. For comparison, POYO-mp trained on 100 sessions (that is, POYO-mp was trained on more data) achieved $R^2$ of 0.9675 $\pm$ 0.0079 by fine-tuning unit ID and $R^2$ of 0.9708 $\pm$ 0.0116 by fine-tuning the entire model on the held-out sessions. When fine-tuned on the held-out sessions, SPINT could further achieve held-out $R^2$ of 0.88 $\pm$ 0.01. These results show the promise of SPINT on datasets beyond the FALCON Benchmark.
>
> [1] Perich, Matthew G., Juan A. Gallego, and Lee E. Miller. "A neural population mechanism for rapid learning." Neuron 100.4 (2018): 964-976. (https://dandiarchive.org/dandiset/000688?search=000688&pos=1)
>
> **W4:** Due to the character limits, we kindly ask the reviewer to check our Q3 response to Reviewer CKq1.
>
> **W5:** We appreciate the reviewer’s comment on the cross-task trainability. As the reviewer suggested, a modification to the SPINT architecture to employ a readout scheme as in POYO+ will facilitate SPINT’s training across behavior tasks. SPINT could be extended to have this multi-dataset pretrainability in future works. We will add this perspective to the Limitations section in the revised manuscript.
>
> **W6:** We thank the reviewer for bringing up similar dropout schemes employed in previous works that we missed. We will cite the references the reviewer suggested, and revise our wording so as to not overclaim dynamic dropout as our main contribution. One relatively minor difference between SPINT’s dropout scheme and that of earlier implementations is that dynamic dropout uses the full valid interval of probabilities in SPINT (i.e., [0,1]), instead of a limited low-probability range (i.e., [0, 0.2]). In Figure 4B, by “without dynamic dropout” we meant with no dropout at all. Due to the character limits, we kindly refer the reviewer to the dynamic dropout table in W2 response to Reviewer CKq1 for further dropout analyses.
>
> **Q1:** Please see the above comments in W6.
>
> **Q2:** Although we designed SPINT for few-shot unsupervised adaptation, SPINT could also be used for few-shot supervised learning, as the reviewer correctly suggested. We performed experiments where we either fine-tune the IDEncoder or the whole model using calibration data on the few labeled samples of held-out sessions, and observed that this led to improved cross-session performance. Fine-tuning the IDEncoder obtained held-out $R^2$ of 0.69 $\pm$ 0.10, and fine-tuning all model weights obtained held-out $R^2$ of 0.70 $\pm$ 0.10, as compared to the gradient-free unsupervised SPINT with held-out $R^2$ of 0.64 $\pm$ 0.10. We will report the new results in the revised manuscript.
>
> **Q3:** Intended to be lightweight, our SPINT model does not have self-attention layers and only has 1 cross-attention layer for simple information aggregation. To analyze the scaling effect of model size, we provide the results for two sets of experiments: 1) similar to POYO, we added self-attention layers before the cross-attention layers, with the number of self-attention layers ranging from 1 to 4. 2) We scaled up the cross-attention layers from 1 to 5. Our results are listed below.
>
> | Number of self-attention layers | 0 (Ours)  | 1 | 2 | 3 | 4 |
> |:--:|-|-|-|-|-|
> |      $R^2$     | 0.64$\pm$0.10 | 0.63$\pm$0.13 | 0.57$\pm$0.13 | 0.61$\pm$0.10 | 0.60$\pm$0.15 |
>
> | Number of cross-attention layers | 1 (Ours)  | 2| 3 | 4 | 5 |
> |:--:|-|-|-|-|-|
> |      $R^2$     | 0.64$\pm$0.10 | 0.65$\pm$0.10 | 0.65$\pm$0.10 | 0.64$\pm$0.11 | 0.62$\pm$0.13 |
>
> Our results indicate that scaling up model size by adding self-attention and cross-attention layers does not significantly improve the performance. This might be due to the small scale of the FALCON datasets, as the reviewer noted. We will incorporate these results and the associated interpretation into the revised manuscript.
>
> **Q4:** Our single behavior query was initialized at random and learned during training. The behavior query Q has dimension BxW, where B is the number of behavior covariates and W is the neural window length. For multi-task extension, there can be multiple queries for decoding behavior in the corresponding task.
>
> **Q5:** We agree with the reviewer that the individual spike tokenization in POYO could potentially work better since spike timings contain helpful information that can be lost in the binning process. However, using POYO spike tokenization in the context of the FALCON continuous decoding paradigm is difficult, as the benchmark strictly requires participating models to adopt the binned spike format for evaluation. Please also see our additional comments in W2 and W3.
>
> **Q6:** We performed a hyperparameter search to find the reported optimal sliding window size. Heterogeneity in the task structure leads to different optimal sizes for each task, as a trial in H1 typically lasts for ~20s, much longer than trials in M1 and M2. Due to the character limits, we refer to the window size table in Q2 from Reviewer fpRR.

---

> > ### Comment · Reviewer_MJ6k · 2025-08-02
> >
> > Thank you very much for your response, I appreciate your efforts in addressing my points. Also, apologies for the delay between my acknowledgment and comment. Some points:
> > * It was not immediately clear to me that the focus here was to isolate the problem of cross-session non-stationarity – to me it seemed like a method proposed as another general approach with good few-shot/transfer properties. I would appreciate if the authors could make this crystal clear in the abstract and introduction. I appreciate and agree with Reviewer Fnbe's point here.
> > * It is not surprising to me that there is no out-of-the-box transfer, and also not surprising that full finetuning works. I thank the authors for their response and experiment on this. I don't think the experiment necessarily shows the subject transferability of SPINT itself (where the focus is on few-shot label-free calibration), but rather just that pretrained models transfer better when fully finetuned.
> > * The intention behind mentioning POYO-style models was solely to understand what approaches work best in which regime through comparisons. I thank you for the experiment in this regard, I do think SPINT would perhaps do better if there were more parameters (e.g., self-attention) that bring it closer to POYO. It is also possible that SPINT would get a small boost in performance if it could process individual spikes (with their raw spike-times as in POYO with Perich et al. data). As suggested by Reviewer Fnbe and also mentioned in your response, perhaps a note detailing key similarities and differences between SPINT and NDT- or POYO-style models would make things clearer.
> > * **Some corrections, however**: POYO-style models _do_ support binned spikes as inputs, in addition to a unit/channel embedding (the definition of _unit_ is flexible), you would additionally have a value embedding as well. This is similar to what POYO+ does and is explored in the POSSM paper as well, so I would ask the authors to take note of this when detailing the similarities and differences.
> > * With regards to dynamic dropout, I appreciate your comments and experiment in response to Reviewer CKq1. The clear advantage you have is that you needn't tune the dropout rate and can get comparable performance to the best standard dropout models. You could state this as a clear advantage of dynamic dropout. It doesn't immediately seem like the performance improvement is statistically significant, and I think it would have been important to compare dynamic dropout with standard dropout in addition to no dropout. The new experiment is something I appreciate as it elucidates your robustness to hyperparameter choices (or lack thereof).
> > * The IDEncoder having to learn the within-trial dynamics is a little concerning and may not work at scale for a multi-subject model (decoupling them could be an interesting extension for future work).
> > * Thanks for these additional experiments on full finetuning and adding self-attention. Your intuition seems reasonable.
> > * Long context performance might depend on the specific position embeddings you use and how many parameters your output cross-attention has, just a note.
> >
> > I've read all the rebuttal responses from the authors, and overall, I'm happy to raise my score. Thank you for your time and effort once again!

---

> > > ### Author Response · Authors · 2025-08-03
> > >
> > > We thank the reviewer for taking time to carefully review our manuscript and responses! We appreciate the constructive comments which greatly helped improve our paper! We are glad that the reviewer took our new experiments into account to raise the score, and will incorporate the reviewer’s points in the revised version of our manuscript :)

---

### Official Review · Reviewer_fpRR · 2025-07-06

**Clarity:** 3
**Significance:** 2
**Originality:** 3
**Rating:** 4
**Confidence:** 3

**Summary:**

The authors propose a framework to address the variability and ordering in the placement of electrodes in the context of real-time, cross-session iBCI decoding and develop a spatial-permutation invariant neural transformer for decoding motor behavior from intracortical spikes. Spike windows are turned into tokens and aggregated via cross-attention.To compensate for the loss of positional information a neural ID encoder derives context-dependent embeddings from a handful of unlabeled "calibration" trials collected at the start of each new session. Next these embeddings are regularized by channel-dropout that removes an arbitrary fraction of units during training. The method has state of the art performance in three multi-day datasets from the FALCON benchmark. It also maintains real time inference without test-time gradients.

**Questions:**

1. Could contrastive or predictive objectives remove the need for labelled data for the neural ID encoder?
2. Would a lightweight temporal transformer on top of the cross-unit attention benefit decoding behaviors with longer latencies?

**Ethical Concerns:**

["NO or VERY MINOR ethics concerns only"]

**Final Justification:**

After reading the authors' rebuttal to my questions and the other reviewers' questions I conclude that there are two important weaknesses in the paper. First, the setting is restrictive especially the fact that it requires calibration data that are session-specific hinders real world applications. Also, the architecture itself needs a lot of tuning to find windows that are able model long-range dependencies since there are no increasing receptive fields. Such validation parameter would require even more data! This last weakness is not apparent in the restricted experimentation of the paper since the windows have already been tuned. I will keep my score as is.

**Limitations:**

yes

**Quality:**

2

**Strengths And Weaknesses:**

Strengths:
1. Permutation invariant and unit agnostic pipeline: Removes the rigid channel ordering assumption enabling training and testing on different population of different size and composition.
2. Context-dependent neural id embeddings.
3. light weight , low-latency architecture: This is particularly significant for closed loop BCI.
4. Consistently outperforms previous approaches on FALCON without test-time gradients.

Weaknesses:
1. Shallow architecture limits long range modeling. Moreover, Neural ID Encoder completely forgets the within-trial structure.
2. Restricted evaluation to 1 human and 2 monkey and behavio limited to arm/hand tasks
3. Gradient -free adaptation still needs session specific calibration data.

---

> ### Author Rebuttal · Authors · 2025-07-31
>
> We thank the reviewer for a thoughtful review and valuable feedbacks. We provide point by point clarifications and answer questions below.
>
> **W1. re:** shallow architecture limiting long range modeling, neural ID Encoder forgetting the within-trial structure.
>
> We understand the reviewer’s concern regarding the long range modeling with shallow architecture. We would like to clarify a point that might have been a source of confusion: the IDEncoder uses the whole trial as input, as opposed to the sliding context window that the cross-attention module uses. Therefore the IDEncoder would be able to learn the within-trial structures.
>
> Secondly, the shallow architecture of SPINT does not necessarily limit long range performance. The opposite might be true for CNNs where each layer has limited receptive field; however, in SPINT, fully connected layers and attention layers are used to model both temporal and spatial population responses, and each temporal step (or spatial unit) is accessible to all other timesteps (or spatial units) regardless of how far apart they are in the context. As long as the sliding window is long enough to include sufficient context for behavior decoding, the model should perform well as we showed in our analyses.
>
> **W2. re:** restricted evaluation to 1 human and 2 monkeys, behavior limited to arm/hand tasks.
>
> We appreciate the reviewer's feedback regarding the number of subjects and diversity of the tasks.
> We provided additional results demonstrating the benefits of using SPINT in cross-subject scenarios (please see table 1 & 2 in response to reviewer MJ6k). The focus of the  hand/arm tasks are mostly based on the availability of the human/monkey experiments. Due to the limitation of recording devices, most neural recordings are collected while monkeys are sitting, leading to the popularity of arm/hand tasks in the literature.
>
> **W3. re:** gradient-free adaptation still needs session specific calibration data.
>
> Indeed, our proposed SPINT framework still requires few session-specific calibration trials although it does not need labels for those trials (thus appreciably improving its practicality). Compared to current state-of-the-art (SOTA) few-shot decoding approaches (NoMAD and CycleGAN) which rely on fine-tuning to achieve desirable motor decoding performance, SPINT is a gradient-free approach that removes the need for test-time fine-tuning.
>
> More broadly, an ideal BCI decoder needs to work under the zero-shot operation regime, meaning that no calibration data and no model fine-tuning are needed. Our approach represents a practicable and effective step towards this goal via a minimal use of calibration trials (in M1 dataset we could achieve 0.65 held-out $R^2$ even with 1 calibration trial, outperforming other SOTA methods).
>
> **Q1. re:** whether contrastive or predictive objectives could remove the need for labelled data for the neural ID encoder
>
> We thank the reviewer for bringing up this insightful idea. We did briefly consider using unsupervised learning objectives such as contrastive or predictive loss to learn the representation for each neural unit. We found that these self-supervised objectives have difficulty in capturing the behavior-relevant aspects of neuronal identity, resulting in performance loss. Additionally, contrastive and predictive objectives require pretraining which necessitates additional training steps on downstream behavior decoding tasks. Thus we opted for the current SPINT design instead of using the contrastive or predictive objectives. We will add this discussion to the revised version of the paper.
>
> **Q2. re:** whether a lightweight temporal transformer on top of the cross-unit attention would benefit decoding behaviors with longer latencies
>
> We thank the reviewer for this suggestion. The ability to model long-range information of SPINT depends on the length of the sliding context window. We performed sweeping experiments to find the best window length that can yield the best performance and found that longer context does not always translate to better performance.
>
> | Window length | 50           | 100     | 200           | 400           | 600           |
> |-------------|---------------|----------------|---------------|---------------|---------------|
> | $R^2$    | 0.65$\pm$0.10 | 0.64$\pm$0.10  | 0.64$\pm$0.10 | 0.60$\pm$0.10 | 0.61$\pm$0.09 |
>
> Table 1: SPINT held-out performance across different window lengths.  Results are reported as mean $\pm$ standard deviation across held-out M1 sessions.
>
> We will incorporate these results and the discussion in the revised version of the paper.

---

> > ### Comment · Reviewer_fpRR · 2025-08-08
> > **Reply to Author Rebuttal**
> >
> > I would like to thank the authors for providing elaborate answers to my questions. I think some of the discussion and ablations especially w.r.t. to the long range context and w.r.t using contrastive and predictive objectives should go to the main paper as it would strengthen the quality of the arguments. I will keep my score.

---

> > > ### Author Response · Authors · 2025-08-09
> > >
> > > We thank the reviewer for the feedback and the acknowledgment of our work. We will include the additional discussions and results during the rebuttal period in the final version of the paper :)

---

### Decision · Program_Chairs · 2025-09-17

**Decision:**

Accept (poster)

**Comment:**

This paper introduces SPINT, a transformer-based framework designed to address the challenge of changing neural populations in long-term intracortical Brain-Computer Interfaces (iBCIs). The model's primary contribution is its ability to operate on unordered sets of neurons, using a novel positional embedding to dynamically infer neural identities, which allows for robust generalization across different recording sessions. The authors demonstrate that SPINT outperforms existing methods and uniquely enables few-shot, gradient-free adaptation, eliminating the need for computationally expensive alignment or fine-tuning during deployment.

The reviewers were in strong agreement about the paper's positive contributions, consistently praising its core idea as novel, clever, and highly relevant to the BCI field. They found the proposed permutation-invariant transformer, which treats neurons as an unordered set to handle cross-session drift, to be a sound and well-designed solution. The reviewers also positively noted the model's strong performance, highlighting that it outperformed baselines on a standard benchmark. They especially appreciated the practical aspects of the method, such as its lightweight architecture and its efficient, gradient-free adaptation scheme, which they saw as a significant advantage for real-world applications. The clarity of the writing and the overall quality of the presentation were also common points of positive assessment.

Despite the positive reception of the core concept, the reviewers also converged on several critiques, primarily concerning the limited scope of the evaluation. They consistently pointed out that the experiments were restricted to only three single-subject datasets and were focused exclusively on motor decoding tasks. This narrow evaluation raised questions about the model's generalizability to new subjects, different tasks, and more complex in-vivo conditions. Another key point of agreement was that while the adaptation method was innovative, the necessity of collecting new, session-specific calibration data for each use was a practical limitation that tempered the model's "plug-and-play" claims. Some reviewers also felt the paper could have been strengthened by a more thorough comparison to other similar architectures and a deeper exploration of the model's failure cases.

Nonetheless, with the overall positive reviews, and scores post-rebuttal, a decision of accept (poster) was reached.